# Prevalence of bacterial coinfection and patterns of antibiotics prescribing in patients with COVID-19: A systematic review and meta-analysis

Faisal Salman Alshaikh[1,2]*, Brian Godman[1,3,4]*, Oula Nawaf Sindi[1,5], R. Andrew Seaton[6,7], Amanj Kurdi[1,4,8,9]

1 Strathclyde Institute of Pharmacy and Biomedical Science (SIPBS), University of Strathclyde, Glasgow, United Kingdom, 2 Pharmaceutical Services, Bahrain Defence Force Military Hospital, Riffa, Kingdom of Bahrain, 3 Division of Public Health Pharmacy and Management, School of Pharmacy, Sefako Makgatho Health Sciences University, Pretoria, South Africa, 4 Centre of Medical and Bio-allied Health Sciences Research, Ajman University, Ajman, United Arab Emirates, 5 Pharmaceutical Sciences Department, Fakeeh College for Medical Sciences, Jeddah, Kingdom of Saudi Arabia, 6 Queen Elizabeth University Hospital, Glasgow, United Kingdom, 7 Scottish Antimicrobial Prescribing Group, Healthcare Improvement Scotland, Glasgow, United Kingdom, 8 Center of Research and Strategic Studies, Lebanese French University, Erbil, Kurdistan Region Government, Iraq, 9 Department of Pharmacology and Toxicology, College of Pharmacy, Hawler Medical University, Erbil, Kurdistan Region Government, Iraq

* ph.faisalalshaikh@gmail.com (FSA); brian.godman@smu.ac.za (BG)

**Data Availability Statement:** All relevant data are within the paper and its Supporting Information files.

## Abstract

### Background

Evidence around prevalence of bacterial coinfection and pattern of antibiotic use in COVID-19 is controversial although high prevalence rates of bacterial coinfection have been reported in previous similar global viral respiratory pandemics. Early data on the prevalence of antibiotic prescribing in COVID-19 indicates conflicting low and high prevalence of antibiotic prescribing which challenges antimicrobial stewardship programmes and increases risk of antimicrobial resistance (AMR).

### Aim

To determine current prevalence of bacterial coinfection and antibiotic prescribing in COVID-19 patients.

### Data source

OVID MEDLINE, OVID EMBASE, Cochrane and MedRxiv between January 2020 and June 2021.

### Study eligibility

English language studies of laboratory-confirmed COVID-19 patients which reported (a) prevalence of bacterial coinfection and/or (b) prevalence of antibiotic prescribing with no restrictions to study designs or healthcare setting.

**Funding:** The author(s) received no specific funding for this work.

**Competing interests:** The authors have declared that no competing interests exist.

### Participants

Adults (aged ≥ 18 years) with RT-PCR confirmed diagnosis of COVID-19, regardless of study setting.

### Methods

Systematic review and meta-analysis. Proportion (prevalence) data was pooled using random effects meta-analysis approach; and stratified based on region and study design.

### Results

A total of 1058 studies were screened, of which 22, hospital-based studies were eligible, compromising 76,176 of COVID-19 patients. Pooled estimates for the prevalence of bacterial co-infection and antibiotic use were *5.62*% (95% CI 2.26–10.31) and 61.77% (CI 50.95–70.90), respectively. Sub-group analysis by region demonstrated that bacterial co-infection was more prevalent in North American studies (7.89%, 95% CI 3.30–14.18).

### Conclusion

Prevalence of bacterial coinfection in COVID-19 is low, yet prevalence of antibiotic prescribing is high, indicating the need for targeted COVID-19 antimicrobial stewardship initiatives to reduce the global threat of AMR.

## 1 Introduction

The first case of coronavirus disease 2019 (COVID-19) was reported in December 2019 [1, 2]. Since its emergence, the novel severe acute respiratory coronavirus 2 (SARS-CoV-2) has resulted in a global pandemic. As of January 14th 2022, a total of 318 million confirmed cases have been reported, with 5.5 million confirmed deaths [3]. The presence of bacterial co-infection in COVID-19 has been a widespread concern amongst healthcare professionals due to overlapping clinical features with bacterial pneumonia [4], and the increased risk of morbidity and mortality associated with bacterial co-infections [5]. Presence of bacterial co-infection had been observed during previous viral pandemics including the 1918 influenza pandemic and the 2009 influenza A (H1N1) pandemic [6, 7], with *S. pneumoniae*, *β-hemolytic streptococci*, *H. influenzae*, and *S. aureus*, being the most common causative pathogens of respiratory tract infections [8]. During winter months influenza-associated bacterial infections may account for up to 30% of community acquired pneumonia cases (CAP) [9]. Nevertheless, other respiratory viruses such as Middle East respiratory syndrome coronavirus (MERS-CoV) and SARS-CoV-1 have reported a very low prevalence of bacterial co-infection amongst infected patients [10, 11] potentially attributable to the comparatively small number of cases reported [12].

Concerns regarding bacterial co-infection in patients with COVID-19has led to widespread use of antibiotics empirically in both hospital and community settings [13–17]. The significant increase in antibiotic prescribing during the pandemic challenges antimicrobial stewardship programmes and risks emergence of multi-drug resistant bacteria [18–20], with their associated impact on morbidity, mortality and costs [21–24].

Prior meta-analyses suggest a bacterial coinfection prevalence of <4% - 8% in patients with COVID-19, nonetheless, these studies included a small number of patients [4, 25–27]. The

prevalence of antibiotic prescribing in patients with COVID-19 was 74.6%, reported in a prior meta-analysis, which included literature mostly from Asia [28]. Consequently, this review aims at building on these publications through identifying the prevalence of bacterial co-infection, and the prevalence of antibiotic use in patients with COVID-19 across multiple countries and regions to guide future prescribing. This includes reducing the inappropriate use of antimicrobials during the COVID-19 pandemic where inappropriate use is a potential driver of antimicrobial resistance (AMR) [19, 20, 29].

## 2 Method

### Search strategy

Electronic databases were systematically searched for published literature reporting bacterial coinfection and/or antibiotic use in patients with COVID-19. The databases searched included OVID MEDLINE, OVID EMBASE, Cochrane library and MedRxiv, with articles published between December 2019 and 29th June 2021. The search terms and keywords used included terms related to "COVID-19", "Coinfections" and "Antibiotics" (**See *S1 Data*)**. The results of the search conducted were imported into Covidence online software for systematic reviews, in which duplicate publications were removed. Reporting was based on the Preferred Reporting Items for Systematic Reviews and Meta-Analyses (PRISMA) guidelines for systematic reviews. The study protocol was registered in the international register of systematic reviews, PROSPERO, under the following ID: CRD42021261734.

### Study selection

Two reviewers (FA and ON) independently screened tittles and abstracts and read full texts to assess if they met the pre-set inclusion criteria, disputes were settled by third a reviewer (AK). All English language articles, irrespective of their primary outcomes, reporting bacterial coinfection rate and/or antibiotics use in, laboratory-confirmed (via Reverse transcription polymerase chain reaction (RT- PCR)), COVID-19 human adult patients ($\geq$ 18 years) in all healthcare settings were included (Outpatients and Inpatients), neonates/children population were excluded due to potentially low prevalence of COVID-19 in the population during early waves of the pandemic with low morbidity and mortality. Studies in which patients with suspected COVID-19, based on clinical symptoms and not laboratory confirmed RT-PCR, were excluded. No restrictions to study design were applied. Case reports, case notes, editorials, letters, systematic review, meta-analysis and qualitative studies were excluded. Abstract only publications with no full text were also excluded.

Non-peer reviewed/ Pre-prints publications on MedRxiv were also included if the papers contained relevant information regarding the topic of interest. This approach of including non-peer reviewed in such meta-analysis during the COVID-19 pandemic had become common place, however, our rationale for this is to try to include larger number of patients in the meta-analysis, to address our outcome of interest.

### Data extraction and quality assessment

Data was extracted into a standardised collection form that was created using Microsoft Excel 2016, by reviewers FA and ON. Data collected for information regarding the demographics of the studies included the following variables: first author; publication year; country of publication; study design (Retrospective, prospective, RCTs etc. . .); is the study multicentre; study setting (Community, hospital, mixed etc. . .); if the study was peer-reviewed; number of positive patients with COVID-19; proportion of male population; and the average age. Data was

collected for the following variables: prevalence of bacterial coinfection (defined as a bacterial coinfection within 48 hours of positive COVID-19 diagnosis and hospital admission), studies looking into super-infection and/or secondary-infection (occurring at 48 hours of hospital admission), were not included; and prevalence of antibiotic use among patients with COVID-19, within first 48 hours of diagnosis. The following information, if reported, was also collected: bacterial species isolated; the prevalence of most common bacteria; most common site of infection of bacterial infection; clinical outcomes of co-infected patients; antibiotic class prescribed; timing of antibiotic initiation in relation to COVID-19 onset and clinical outcomes of patients prescribed antibiotics. The Newcastle-Ottawa Scale (NOS) was used to assess the quality of the observational studies included in the review [30].

### Data synthesis, sensitivity analysis and publication bias

The two primary outcomes were the prevalence of bacterial coinfection in COVID-19 patients and the prevalence of antibiotics use in patients with COVID-19. Further sub-group analysis was conducted based on studies' region/continent and design. Proportion (prevalence) outcome data across all studies were pooled using a random effect meta-analysis with Freeman and Tukey method [31]. Results were presented using forest plots, to demonstrate the studies' effect size, and 95% confidence intervals (CI). Heterogeneity was assessed using $I^2$ statistic. A value below 40% was considered to be low heterogeneity, 30–60% was considered to be moderate heterogeneity, 50–90% was substantial, and 75–100% is considerable heterogeneity [32]. Publication bias was assessed through Funnel plots followed by Egger's asymmetry test [33]. All analyses were carried out using STATA/BE 17.0 for Windows (64-bit x 86–64) using the *Metaprop* command package.

## 3 Results

A total of 1183 studies were identified and 125 duplicates were removed. A total of 1058 studies were screened for title and abstract, 81 were screened by full-text screening and 22 studies were eligible for inclusion in the final analysis [24–45] (*Fig 1*). Prevalence of bacterial coinfections was reported in 20 of the 22 studies included, whilst prevalence of antibiotics use was reported in 18 studies only (*Table 1*).

### Study characteristics

Retrospective cohort studies accounted for the majority of the studies involved (n = 18, 81%), whilst prospective cohort studies accounted for the remaining (n = 4, 18%). Of the 22 studies included, 3 (13%) studies were pre-prints [50, 51, 53], whilst the remaining (n = 19, 86%) were peer-reviewed. A total of 13 (59%) studies were conducted in multicentre settings, whilst the remainder (n = 9, 40%) were conducted in single centre settings. All of the studies included were conducted in hospital settings, whether it be in a normal, isolation or an intensive care ward. Twenty one (95%) out of the 22 studies have been classified as a "Good" rating during the quality assessment process (*Table 2*).

### Geographical distribution

The majority of the studies included in the review took place in the United States of America (USA) (n = 10, 45%), followed by the United Kingdom (UK) (n = 4, 18%), China (n = 3, 14%) and 1 study each in France, Germany, Indonesia, The Netherlands and Spain. Continent-wise, 10 (45%) studies were from North America, 8 (36%) from Europe and 4 (18%) were from Asia.

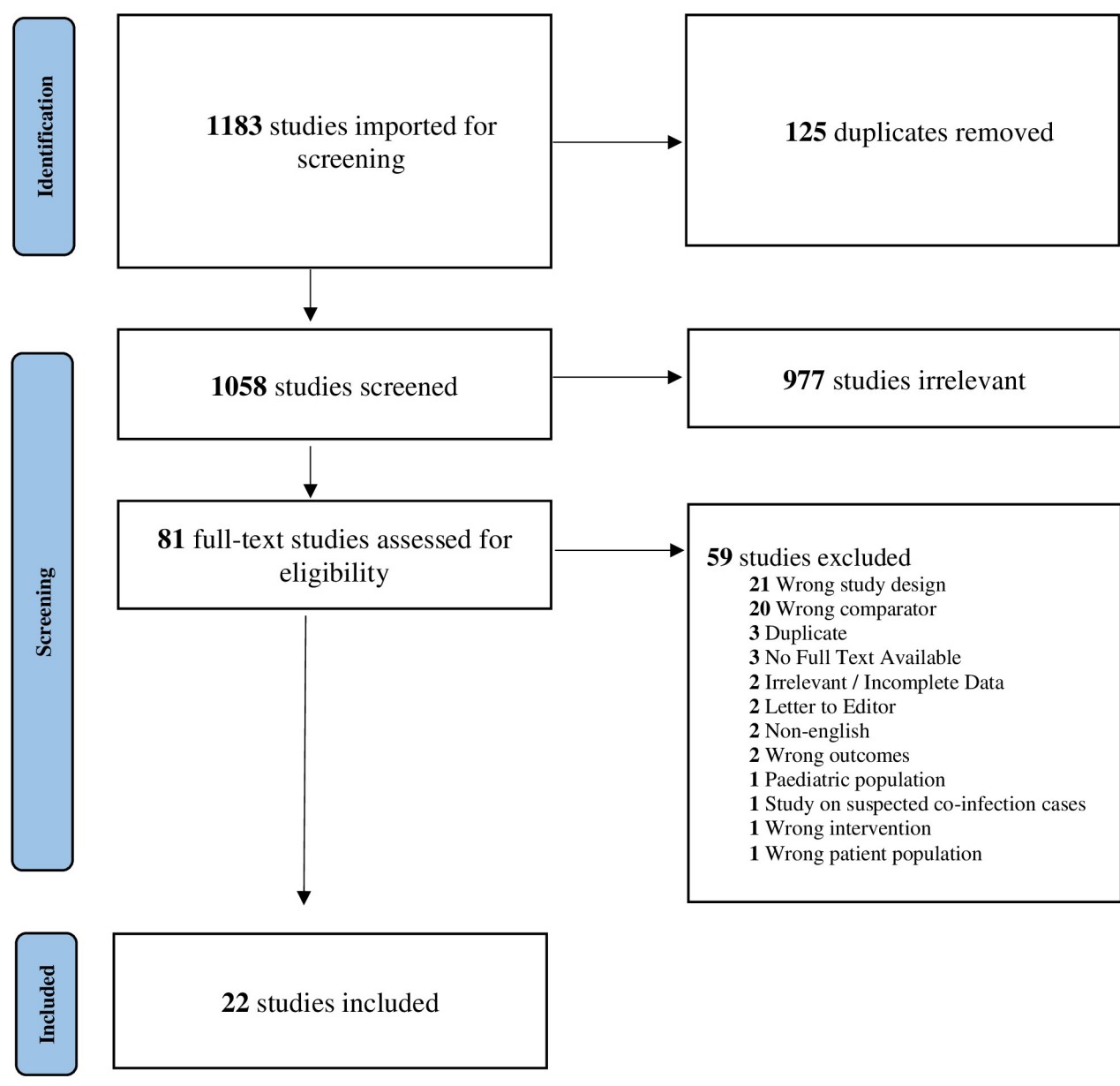

**Fig 1. Study flow diagram based on PRISMA guidelines.**

## Patients characteristics

A total of 76,176 adult patients with RT-PCR confirmed COVID-19 were included from 22 studies, with studies by Russell *et al.* [44] (UK, 48,902 patients) and Puzniak *et al.* [34] (US, 17,003 patients) comprising 86.5% of the overall study population. The mean age of patients, was 61 years (IQR 59 67, range 36–74) and mean proportion of male subjects was 54% (IQR 50–63).

Of all the 20 studies (90%) reporting on bacterial coinfection, the most commonly reported bacterial organism was *S. aureus (*n = 8, 40%), followed by *E.coli* (n = 3, 15%). The most common source of bacterial coinfection was respiratory (n = 10, 50%), followed by blood (n = 2, 10%) and urine (n = 2, 10%).

**Table 1. Summary of study and patients' characteristics.**

| | Author, Year | Country | Study Design | Multicentre? (Y/N) | Study Setting | Peer-reviewed? (Y/N) | Quality Rating | COVID-19 Patients, n | Male, n (%) | Age (SD/IQR) | Bacterial Coinfection, n (%) | Most common Bacteria, N (%) | Antibiotic Use, N (%) | Antibiotic Class (most common) |
|---|---|---|---|---|---|---|---|---|---|---|---|---|---|---|
| 1. | Puzniak L, 2021[34] | USA | Retrospective Cohort | Y | Hospital | Y | Good | 17003 | 9026 (53) | 61.7 (18) | 2889 (16.99) | Enterobacterales, 1594 (55.17) | 11562 (68) | Cephalosporins |
| 2. | Wang L, 2020 [35] | UK | Retrospective Cohort | Y | Hospital | Y | Good | 1396 | 903 (65) | 67.4 (16.2) | 37(2.65) | E. Coli, 6 (16.22) | 36 (2.58) | |
| 3. | Michael S, 2020 [36] | USA | Retrospective Cohort | Y | Hospital | Y | Good | 73 | 35 (48) | | | | 27 (36.99) | NR |
| 4. | S. Hughes, 2020 [37] | UK | Retrospective Cohort | Y | Hospital | Y | Good | 836 | 519 (62) | 69 (55–81) | 27 (3.23) | S. aureus, 4 (14.81 | | |
| 5. | Contou D, 2020 [38] | France | Retrospective Cohort | N | Hospital | Y | Good | 92 | 73 (79) | 61 (55–70) | 26 (28.26) | S. aureus, 10 (38.46) | 39 (42.39) | Cephalosporins |
| 6. | Cheng, L, 2020 [39] | China | Retrospective Cohort | Y | Hospital | Y | Good | 147 | 85 (58) | 36 (24–54) | 4 (2.72) | | 52 (35.37) | Cephalosporins |
| 7. | Neto A G M,2020 [40] | USA | Retrospective Cohort | N | Hospital | Y | Good | 242 | 123 (51) | 66 (14.75) | 46 (19.01) | E. Coli, 12 (26.09) | 162 (66.94) | Cephalosporins |
| 8. | Lardaro T, 2020 [41] | USA | Retrospective Cohort | Y | Hospital | Y | Good | 542 | 269 (50) | 62.8 (16.5) | 20 (3.69) | S. aureus, 7 (35) | | |
| 9. | Chen S, 2020 [42] | China | Retrospective Cohort | N | Hospital | Y | Good | 408 | 196 (48) | 48 (34–60) | 25 (6.13) | mycoplasma pneumonia, 13 (52) | 60 (14.71) | NR |
| 10. | Baskar V, 2021 [43] | UK | Retrospective Cohort | Y | Hospital | Y | Good | 254 | 164 (65) | 59 (49–69) | 14 (5.51) | S. aureus, 4 (28.57) | | |
| 11. | Russell C D, 2021 [44] | UK | Prospective Cohort | Y | Hospital | Y | Good | 48902 | 28116 (58) | 74 (59–84) | 318 (0.65) | E. Coli, 64 (20.13) | 39528 (80.83) | Penicillin/B-lactams |
| 12. | Lehmann C J,2021 [45] | USA | Retrospective Cohort | N | Hospital | Y | Poor | 321 | 155 (48) | 60 (17) | 7 (2.18) | S. aureus, 2 (28.57) | 222 (69.16) | NR |
| 13. | Vaughn V,2021 [46] | USA | Retrospective Cohort | Y | Hospital | Y | Good | 1705 | 885 (52) | 64.7 (53–77) | 59 (3.46) | | 965 (56.6) | Cephalosporins |
| 14. | Miao Q, 2021 [47] | China | Retrospective Cohort | N | Hospital | Y | Good | 323 | | | 17 (5.26) | Klebsiella pneumonia, 11 (64.71) | | |
| 15. | Karami Z, 2020 [48] | Netherlands | Retrospective Cohort | Y | Hospital | Y | Good | 925 | 591 (64) | 70 (59–77) | 7 (0.76) | S. aureus, 4 (57.14) | 556 (60.11) | Cephalosporins |
| 16. | Garcia-Vidal C, 2021 [49] | Spain | Retrospective Cohort | N | Hospital | Y | Good | 989 | 552 (56) | 62 (48–74) | 31 (3.13) | Streptococcus pneumonia, 12 (38.71) | 917 (92.72) | Macrolide |
| 17. | Crotty M P, 2020 [50] | USA | Prospective Cohort | Y | Hospital | N | Good | 289 | | 58.6 (14.4) | 25 (8.65) | S. aureus, 5 (20) | 271 (93.77) | NR |

*(Continued)*

**Table 1.** (Continued)

| | Author, Year | Country | Study Design | Multicentre? (Y/N) | Study Setting | Peer-reviewed? (Y/N) | Quality Rating | COVID-19 Patients, n | Male, n (%) | Age (SD/IQR) | Bacterial Coinfection, n (%) | Most common Bacteria, N (%) | Antibiotic Use, N (%) | Antibiotic Class (most common) |
|---|---|---|---|---|---|---|---|---|---|---|---|---|---|---|
| 18. | Wei W, 2020 [51] | USA | Prospective Cohort | N | Hospital | N | Good | 147 | 87 (59) | 52 | | | 87 (59.18) | Cephalosporins |
| 19. | Karaba S, 2020 [52] | USA | Prospective Cohort | Y | Hospital | Y | Good | 1016 | 543 (53) | 62 (48–74) | 53 (5.22) | | 717 (70.57) | NR |
| 20. | Martin A, 2020 [53] | USA | Retrospective Cohort | Y | Hospital | N | Good | 208 | 105 (51) | 69 (60–80) | 24 (11.54) | *S. aureus*, 5 (20.83) | 172 (82.69) | Cephalosporins |
| 21. | Rothe K, 2021 [54] | Germany | Retrospective Cohort | N | Hospital | Y | Good | 140 | 90 (64) | 63.5 (17–99) | 3 (2.14) | | 116 (82.86) | Penicillin/B-lactams |
| 22. | Asmarawati T P, 2020 [55] | Indonesia | Retrospective Cohort | N | Hospital | Y | Good | 218 | 120 (55) | 52.45 (14.44) | 13 (5.96) | *NR* | 164 (75.23) | Quinolones |

**Table 2. Risk of bias assessment using Newcastle-Ottawa scale (NOS).**

| | Author, Year | Score per Domain | | | Quality Rating |
|---|---|---|---|---|---|
| | | Selection | Comparability | Outcome | |
| 1. | Puzniak L, 2021 | 4 | 2 | 2 | Good |
| 2. | Wang L, 2020 | 4 | 2 | 2 | Good |
| 3. | Michael S, 2020 | 4 | 2 | 2 | Good |
| 4. | S. Hughes, 2020 | 4 | 2 | 2 | Good |
| 5. | Contou D, 2020 | 4 | 2 | 2 | Good |
| 6. | Cheng, L, 2020 | 4 | 2 | 2 | Good |
| 7. | Neto A G M,2020 | 4 | 2 | 2 | Good |
| 8. | Lardaro T, 2020 | 4 | 2 | 2 | Good |
| 9. | Chen S, 2020 | 4 | 2 | 2 | Good |
| 10. | Baskar V, 2021 | 4 | 2 | 2 | Good |
| 11. | Russell C D, 2021 | 4 | 1 | 1 | Poor |
| 12. | Lehmann C J,2021 | 4 | 2 | 2 | Good |
| 13. | Vaughn V,2021 | 4 | 2 | 2 | Good |
| 14. | Miao Q, 2021 | 4 | 2 | 2 | Good |
| 15. | Karami Z, 2020 | 4 | 2 | 2 | Good |
| 16. | Garcia-Vidal C, 2021 | 4 | 2 | 2 | Good |
| 17. | Crotty M P, 2020 | 4 | 2 | 2 | Good |
| 18. | Wei W, 2020 | 4 | 2 | 2 | Good |
| 19. | Karaba S, 2020 | 4 | 2 | 2 | Good |
| 20. | Martin A, 2020 | 4 | 2 | 2 | Good |
| 21. | Rothe K, 2021 | 4 | 2 | 2 | Good |
| 22. | Asmarawati T P, 2020 | 4 | 2 | 2 | Good |

The most commonly used class of antibiotics were the cephalosporins (8 out of 18 studies), with 7 out of 18 of the studies reporting that antimicrobial use was initiated on admission.

## Meta-analysis of prevalence of bacterial coinfection in COVID-19 patients

A total of 20 studies of the 22 studies included in this review, comprising of 75,956 (99.7%) of the overall study population, investigated bacterial co-infection. Of which, only 3,645 (4.7%) patients were reported to have a confirmed diagnosis of bacterial co-infection. The random effects meta-analysis of all combined studies estimated that the prevalence of bacterial coinfection in patients with COVID-19was 5.62% (95% CI 2.26–10.31), with an $I^2$ value of 99.69%, indicating considerable heterogeneity (*Fig 2*), and an estimate of between-study variance Tau$^2$ value of 0.15.

## Meta-analysis of antibiotic use in COVID-19 patients

Antibiotic use was reported in 55,653 of the total 76,176 patients included in this review, with 18 studies (81%) reporting antibiotic use in patients with COVID-19. The random effects meta-analysis of all combined studies have estimated a prevalence of 61.16% (CI 50.95–70.90) of antibiotic prescribing in COVD-19, with an $I^2$ value of 99.77%, indicating considerable heterogeneity (*Fig 3*), and an estimate of between-study variance Tau$^2$ value of 0.19.

## Bacterial coinfection by region

The prevalence of bacterial coinfection was highest in North America (7.89%, 95% CI 3.30–14.18), followed by Asia (5.3%, 95% CI 4.03–6.73), with Europe having the lowest prevalence

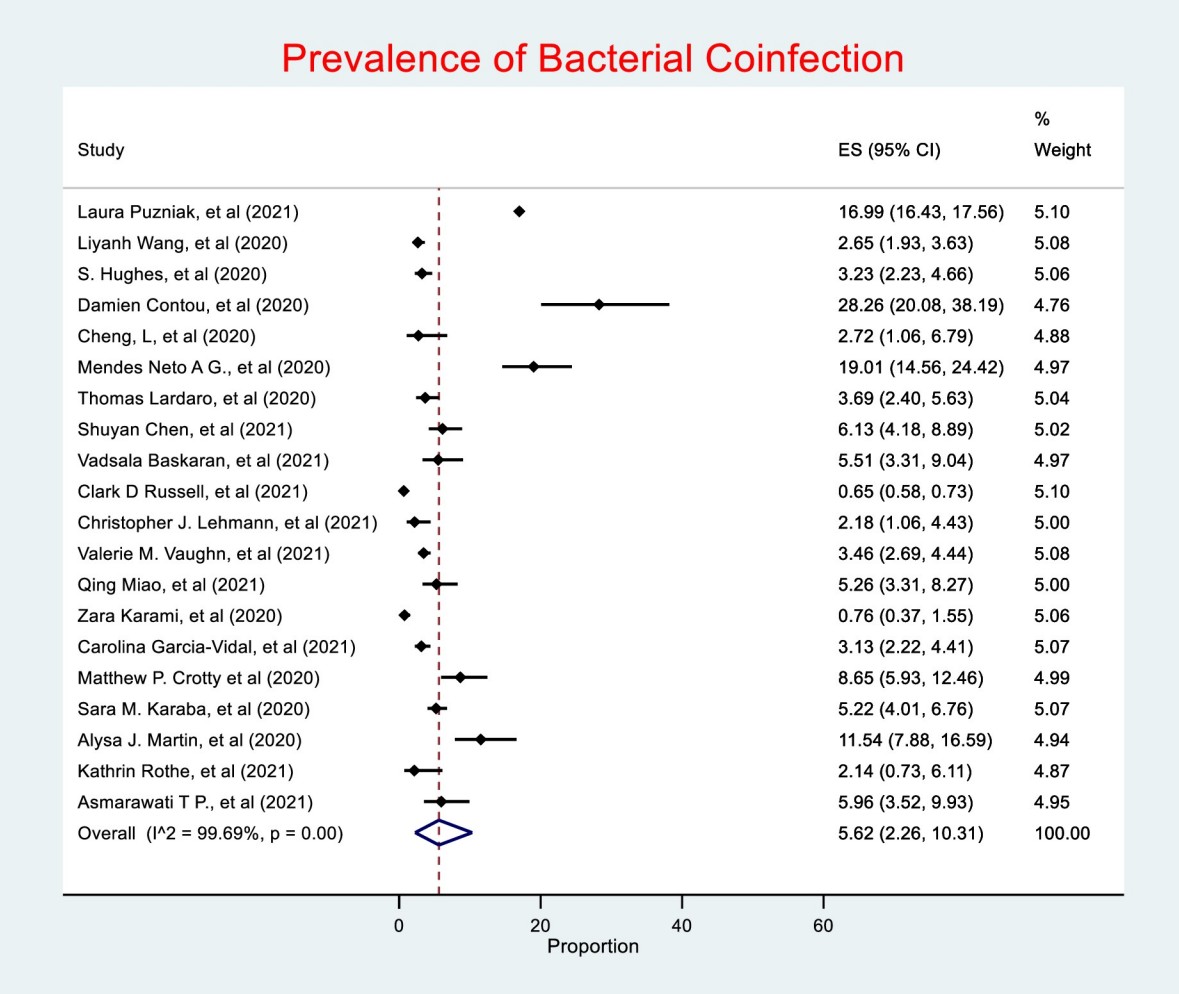

**Fig 2. Prevalence of bacterial coinfection. (ES (effect size), 95% CI (95% confidence interval)).**

(3.57%, 95% CI 1.72–6) (*Fig 4*). Heterogeneity was considerable in both North America and Europe, $I^2$ = 98.89% and 96.75% respectively. Studies in Asia had low heterogeneity with an $I^2$ value of 0%.

## Bacterial coinfection by study design

Retrospective cohort studies had the highest prevalence of bacterial coinfection (5.92%, 95% CI 2.79–10.07), whilst prospective cohort studies had a prevalence of 3.97% (95% CI 0.38, 10.92) (*Fig 5*). Heterogeneity was considerable in both retrospective and prospective, $I^2$ = 98.88% and 98.62% respectively.

## Antibiotic use by region

North America had the highest antibiotic use in patients with COVID-19per region (68.84%, 95% CI 62.27–75.05), followed by Europe (60.01%, 95% CI 25.50–89.67), with Asia having the lowest prevalence of antibiotic use (40.81%, 95% CI 7.75–79.65) (*Fig 6*). Heterogeneity was considerable across all with studies in Europe being the most heterogeneous ($I^2$ = 99.91%), followed by Asia ($I^2$ = 99.18%), followed by North America ($I^2$ = 97.28%).

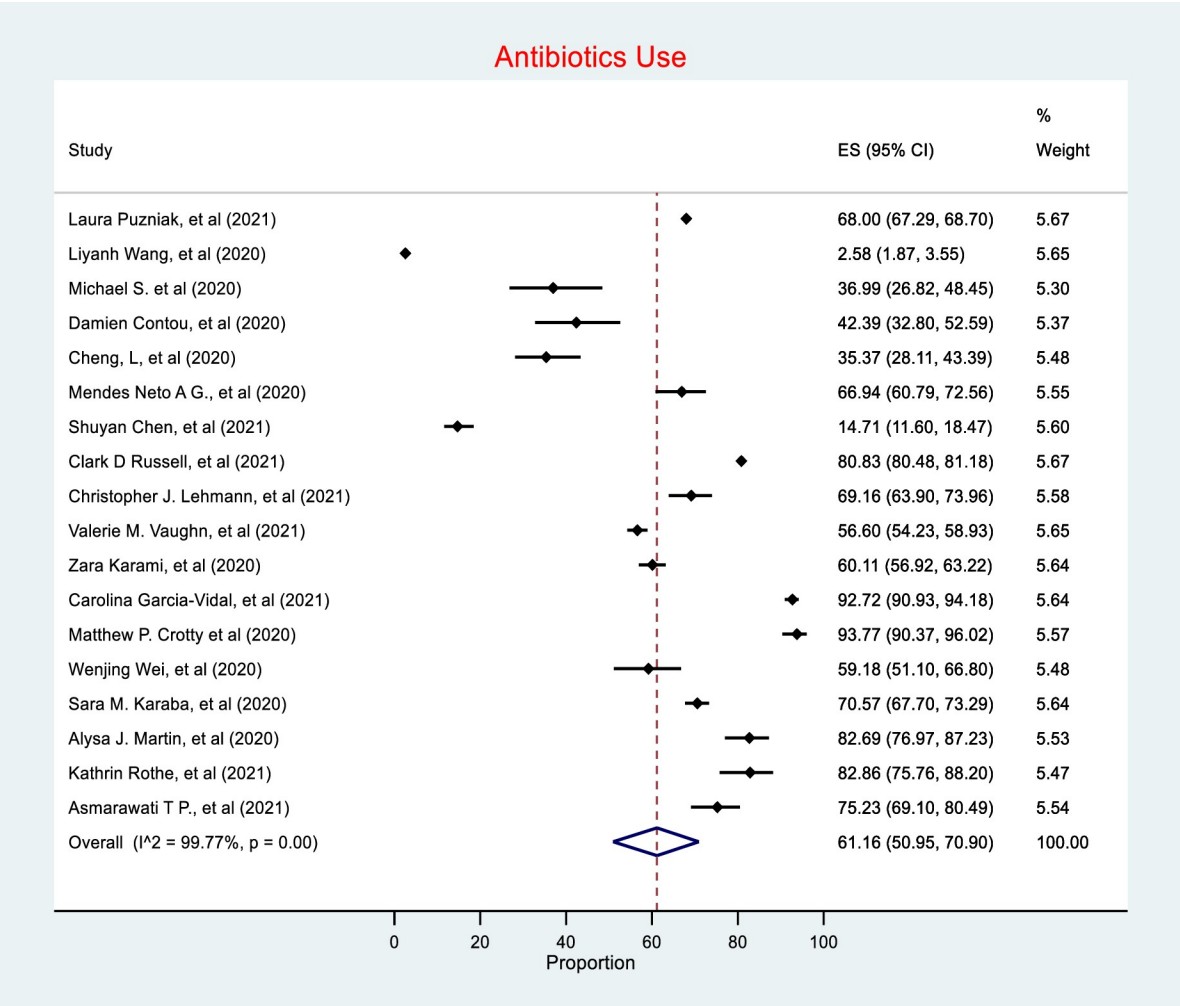

**Fig 3. Antibiotic use.**

### Antibiotic use by study design

Prospective cohort studies had the highest estimate of antibiotic prescribing prevalence (77.83%, 95% CI 68.09–86.23), followed by retrospective cohort studies (56.02%, 95% CI 39.40–71.97) (*Fig 7*). Heterogeneity was considerable in both Retrospective and Prospective cohort studies, with $I^2$ value of 99.72% and 97.82%, respectively.

### Bias assessment (publication bias)

As detected by the funnel plots generated (*Fig 8*), there was no evidence of publication bias. This is further supported by the objective results (p-values) obtained through Egger's asymmetry test for studies in both prevalence of bacterial coinfection and antibiotic use, p-values were 0.43 and 0.59, respectively.

## 4 Discussion

The aim of this systematic review and meta-analysis was to determine the prevalence of bacterial coinfection and antibiotic use in patients with COVID-19. The prevalence of bacterial

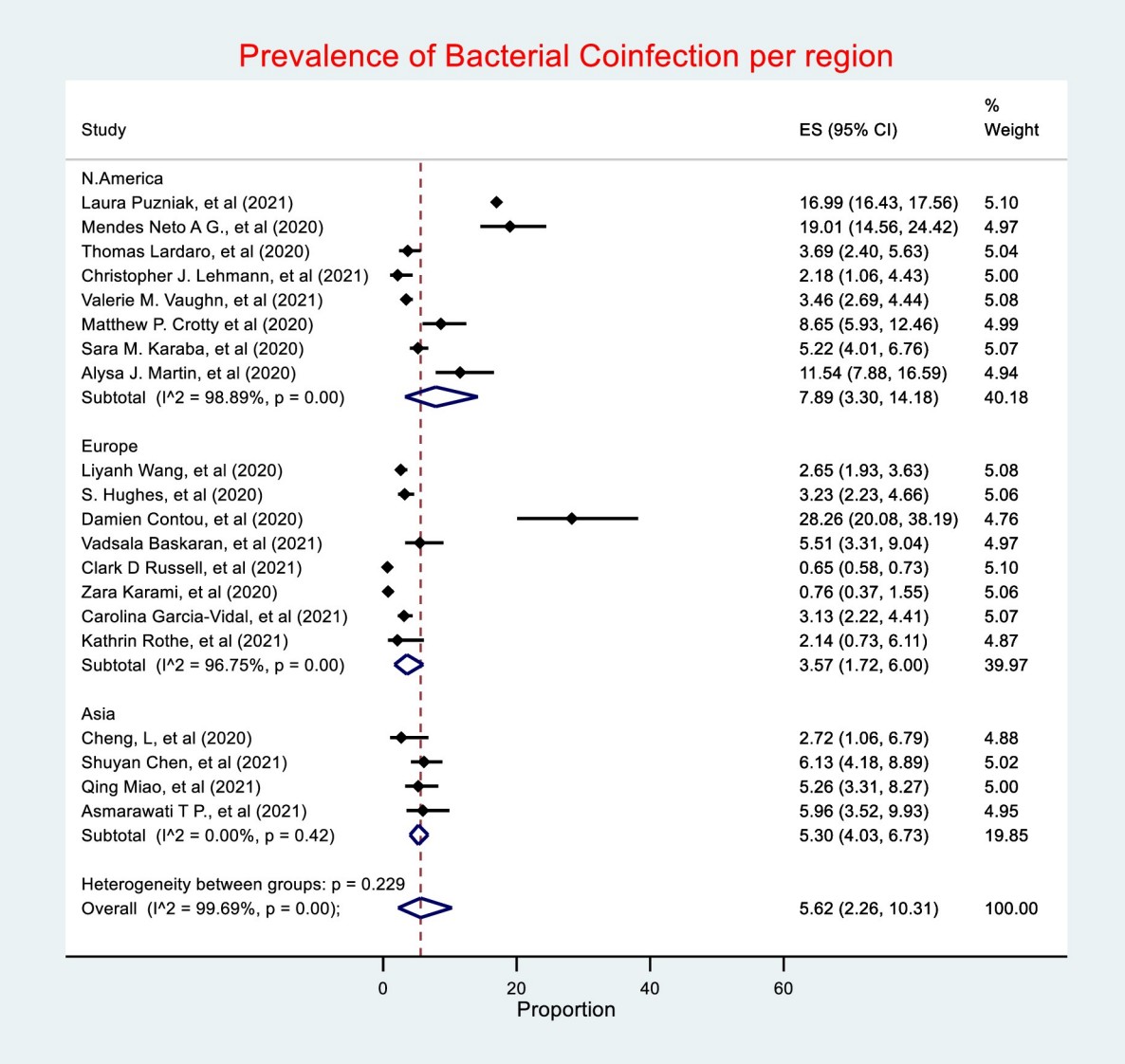

**Fig 4. Prevalence of bacterial coinfection by region.**

coinfection amongst patients with COVID-19was 5.62% (95% CI 2.26–10.31), whilst, the use of antibiotic agents amongst patients with COVID-19 was 61.77% (CI 50.95–70.90). To the best of the authors' knowledge, this is the first meta-analysis to investigate both outcomes at once as well as break the findings down by Region to provide future guidance.

With regards to bacterial coinfection in patients with COVID-19, the findings in this review are consistent with those of previously published studies and smaller systematic reviews addressing this issue (Range <4% - 8%) [4, 25–27]. Bacterial coinfection prevalence was low across all included studies, with the exception to Contou *et al.*, Neto *et al.* and Puzniak *et al.* in which the reported prevalence rates were 28%, 19% and 16% respectively [34, 38, 40].

High prevalence rates reported by Contou *et al.* can be attributed to the study setting, which was the ICU. Symptomatic patients admitted to the ICU were tested for COVID-19 and for bacteriological pathogens afterwards; consequently, potentially reporting higher prevalence of bacterial coinfection. Nonetheless, Contou D *et al.* clearly differentiated in their study

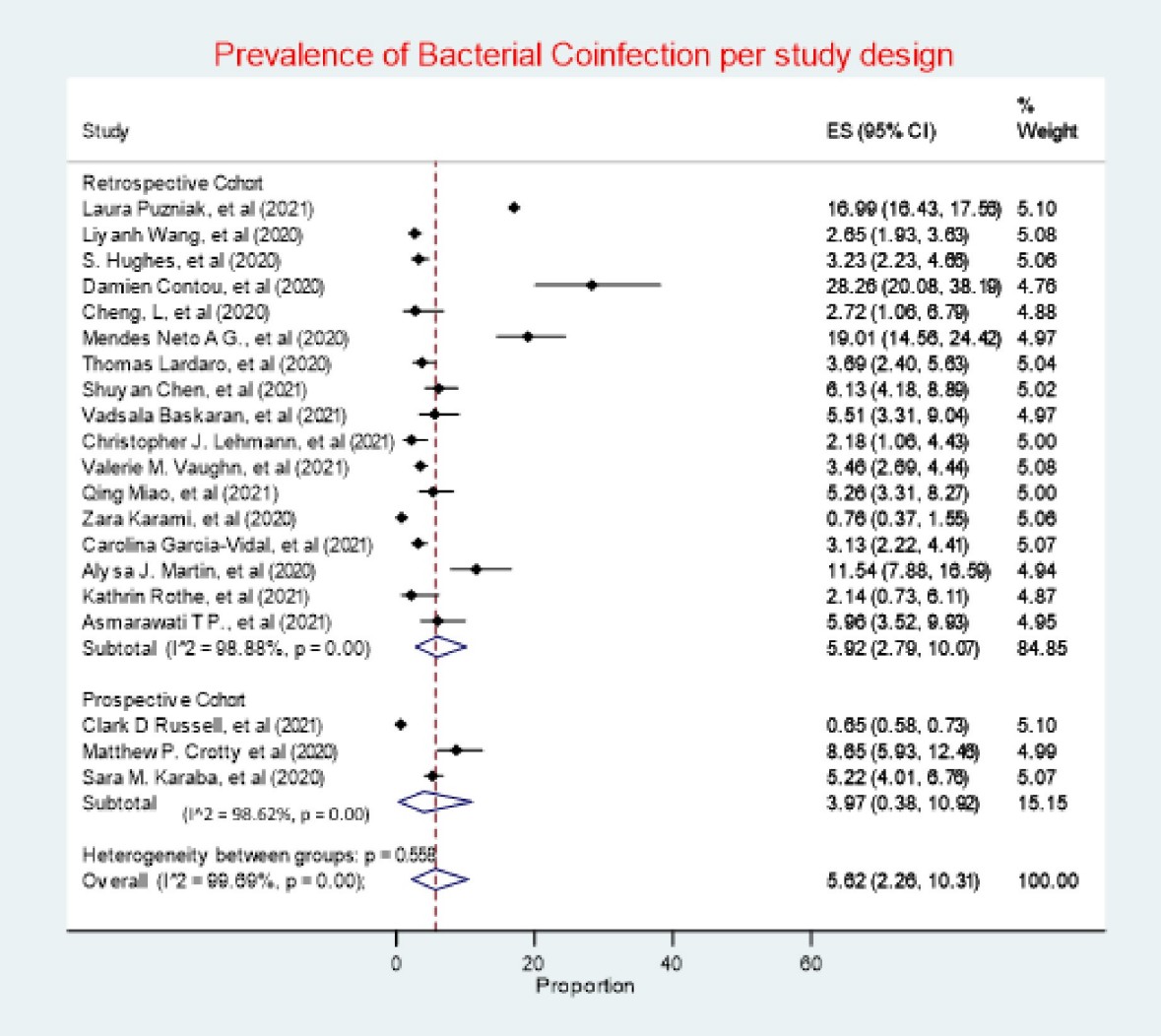

**Fig 5. Prevalence of bacterial coinfection by study design.**

design between coinfections and nosocomial infections. Positive microbiological samples conducted within the first 48 hours of admission were labelled as coinfection, whilst positive microbiological samples after 48 hours were considered to be nosocomial ICU-acquired infections [38].

Urinary tract infections (UTIs) were the most prevalent source of bacterial coinfection (57%) as reported by Neto *et al.* [40]. The authors attributed the high UTI rate to the lack of a fixed defining clinical characteristics of bacterial coinfection and to high risk factors for UTIs amongst the study population, e.g. elderly hospitalised female patients and diabetic patients. It might be surprising that *E coli* was the second most commonly identified organism because *E coli* is an uncommon cause of community acquired pneumonia; this is likely to be driven by the studies including UTI among their coinfections.

High bacterial coinfection prevalence rates (16%) were reported by Puznik *et al.* [34], the second largest study included in this review, when compared to the low prevalence rates reported by Russell *et al.* [44] (0.65%), the largest study in the review. This may be due to a number of factors. These include the frequency of microbiological investigations, in which,

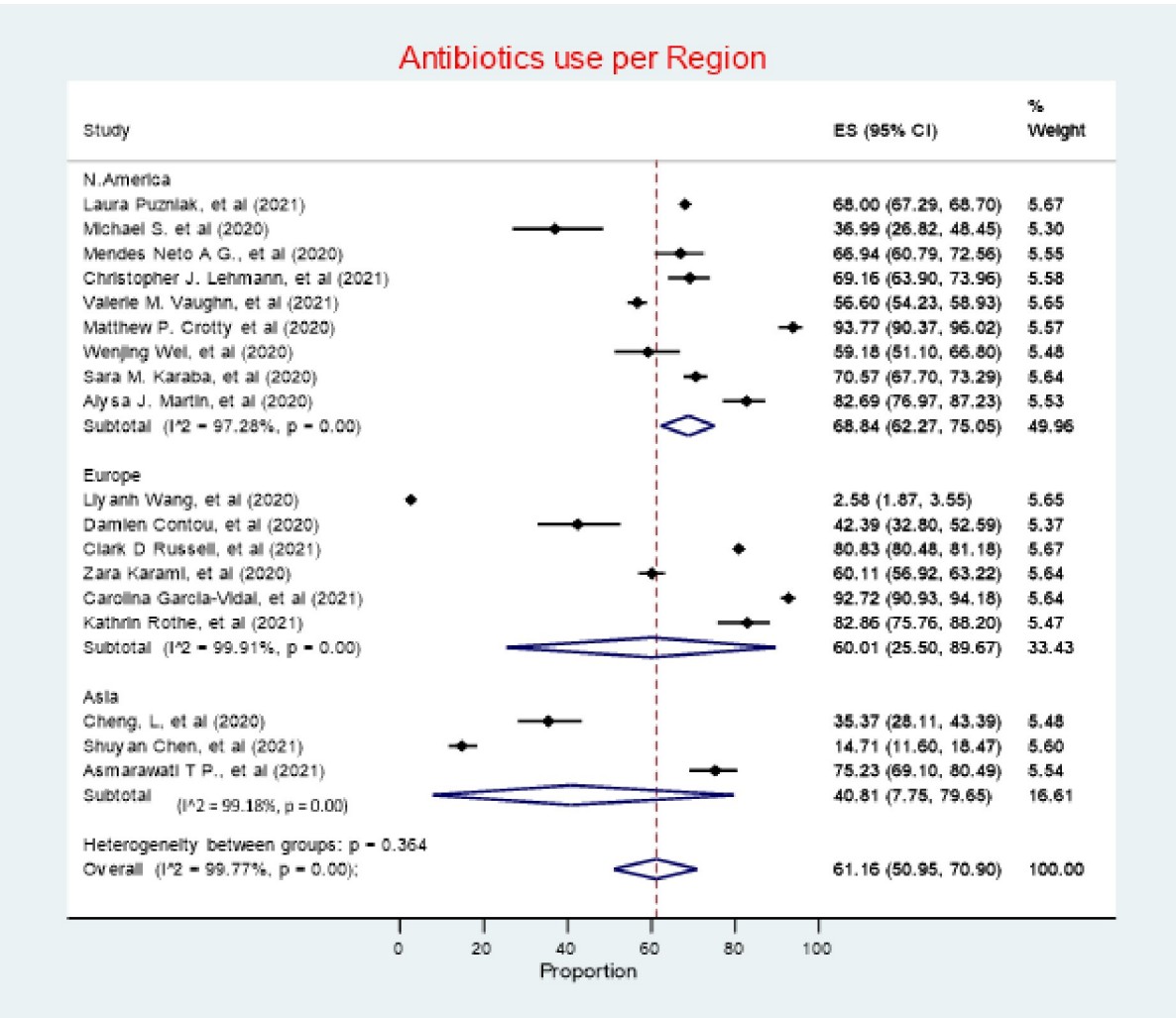

**Fig 6. Antibiotic use by region.**

investigation rates were higher in the study of Puznik *et al*. Interpretation of microbiological results in which gram-negative bacteria in sputum samples of non-ventilated patients were taken which may have over-estimated significance of bacterial coinfection [34].

The analyses conducted around bacterial coinfection in patients with COVID-19 suggests that bacterial coinfection prevalence rates are lower than seen in previous viral pandemics. During the 2009 swine flu pandemic, up to 55% of mortalities were as a result of bacterial pneumonia [46]. Previous pandemics have also reported that *S. pneumoniae, β-hemolytic strep-tococci, H. influenzae*, and *S. aureus* were the most commonly identified bacterial co-pathogens [8]. In this review, *S. aureus* has been the most identified bacterial co-pathogen.

This review also identified very high antibiotic use in patients with COVID-19, which is consistent with previous reviews including those of Langford *et al*. (2021) [28], which reported a prevalence of 74.6% (95% CI 68.3–80.0%). Differences between the results seen in this review and the review of Langford *et al*. may be attributed to the fact that the latter review also included case series with ≥10 patients. This can potentially be attributed to the time period of the pandemic in which the studies were conducted. There was scarceness of cohort studies in

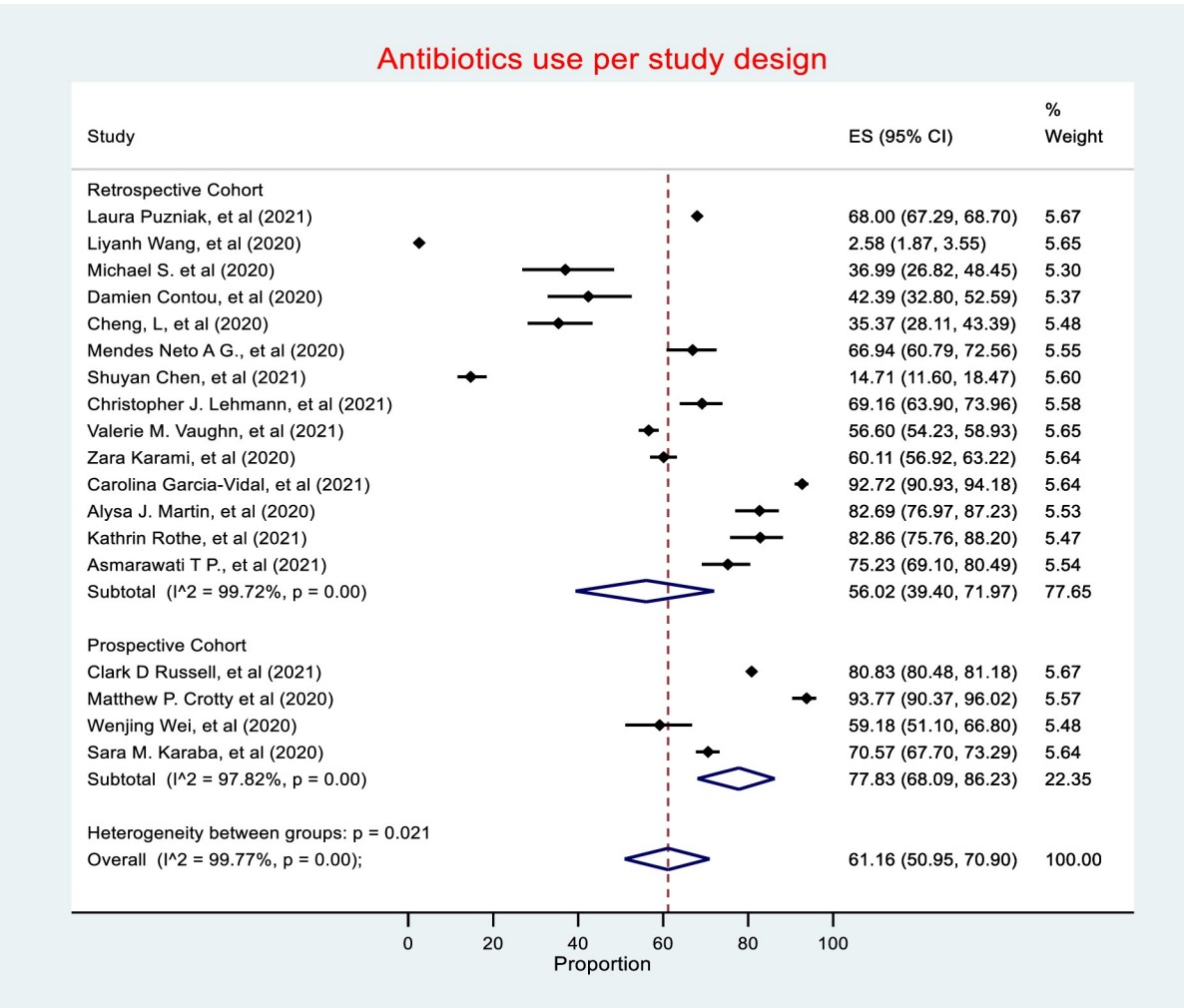

**Fig 7. Antibiotic use by study design.**

the review of Langford *et al.* (2021) [28], which is different to our study. This review also included a wider selection of nations in addition to a larger number of patients.

The increase in antibiotic use observed during this pandemic might have impacted and set-back antimicrobial stewardship (AMS) efforts globally, especially in regions where AMS programmes are just starting as seen in Africa with previous knowledge and resource issues [56–58]. This is starting to change in Africa with a growing number of AMS activities to address identified concerns [59–61]. However, remarkably, in certain regions globally, specifically in Europe, there was a decline in antibiotic use overall in 2020, despite high antibiotic use in COVID-19 positive patients. This can potentially be attributed to a number of factors including social distancing measures and reduction in medical activities [62–64]. Nonetheless, inappropriate use of antibiotics during COVID-19 is a potential driver of the silent AMR pandemic [19, 65]. However, with current changes observed in global human behaviour, relating to personal hygiene, and increased interest in infection control since the emergence of this pandemic, we should see a rise in AMS activities globally [66].

Sub-group analysis based on the key regions demonstrated that the prevalence of reported bacterial coinfection was higher in North America followed by Asia and Europe at 7.89%,

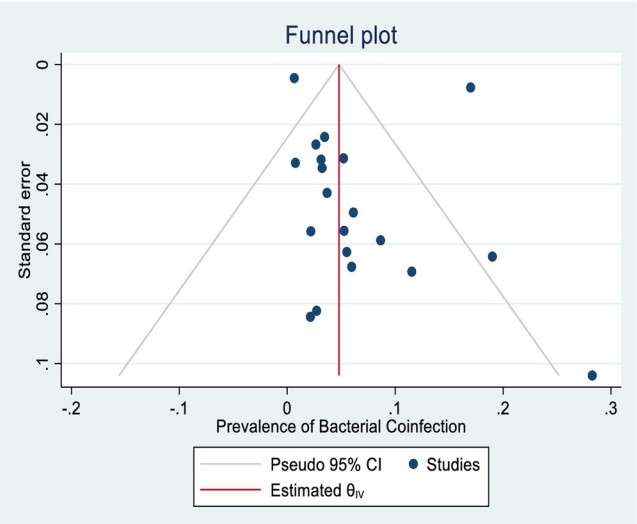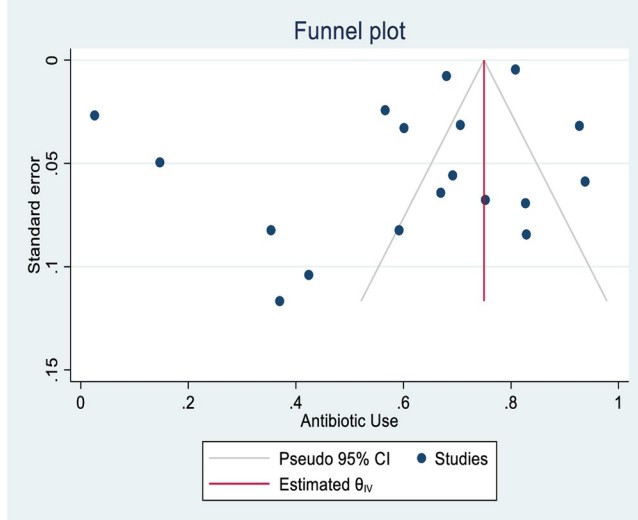

**Fig 8. Funnel plots illustrating the assessment of publication bias for each primary outcome.**

5.30% and 3.57%, respectively. Antibiotic use was also higher in North America (68.84%), followed by Europe (60%) and Asia (40.81%). Our hypothesis suggests that the reason for higher prevalence of bacterial coinfection and antibiotic use in North America is due to the presence of larger number of studies and patients from the region in this review, in addition to possibly higher rates of microbiology investigation and over interpretation of microbiology results. Nevertheless, studies from Asia are reporting high use of antibiotics including the study of Hassan *et al.*, which reported extremely high use of antibiotics (92%) in COVID-19 patients [67], however, this study was not included in our meta-analysis as it has not met our inclusion criteria. We are also aware of more recent studies in Asia reporting high rates since our analysis [17, 68].

In this review, investigating regional distribution of co-infection and antibiotic use was key. Its significance is directly correlated to the fact that antimicrobial use varies considerably across regions, albeit some convergence [69]. It is quite apparent that high antibiotic consumption is common in low- and middle-income countries (LMICs) in contrast to high-income countries (HICs) [69]. In addition, AMR rates vary considerably across countries and regions, with high AMR rates quite evident in regions such as South Asia and Sub-Sahara Africa, therefore, it was practical, in this review, to breakdown antibiotic usage rates by region [70, 71].

In terms of study design, sub-group analysis has demonstrated that retrospective studies had higher prevalence of bacterial coinfection than prospective ones at 5.92% vs 3.97% respectively. Whilst, on the other hand, antibiotic use was higher in prospective than retrospective studies, 77.83% vs 56.02%, respectively. The main hypothesis that might explain these variations in prevalence from the main meta-analyses is the study design itself. Prospective studies had well-defined processes to determine bacterial coinfection in patients with COVID-19, such as pre-defined clinical characteristics that prompt microbiological sampling [44]; hence likely lower bacterial coinfection rates but higher justifiable antibiotic use.

Despite having 10 out of 22 studies included in this review published in 2021, all the studies included have been conducted mainly in the first few months of the pandemic (February and April 2020) with the exception of one study conducted in June 2020 [55]. The results from this review demonstrates that there is insufficient evidence supporting considerable empiric antibiotic prescribing in patients with COVID-19due to a low prevalence of bacterial coinfection.

Nonetheless, antibiotics use was high mirroring the findings in other reviews. As the pandemic evolves, and new COVID-19 specific therapeutics come into clinical practice, it will be important to assess their impact on antibiotic use. The early phase of the pandemic from which most of the published studies to date relate has been characterised by a lack of specific COVID-19 therapies and it may be as treatment options become available, and the understanding of the low prevalence of bacterial co-infection becomes more established, that there will be less reliance or defaulting to antibiotic prescribing. We will be following this up in future studies.

## Strengths and limitations

We believe the key strengths of this review included a comprehensive search strategy spanning several databases, including both pre-prints and peer-reviewed studies, resulting in 22 studies being included, representing over 76,000 patients. In order to overcome any threats to the statistical validity of our pooled estimates (due to the nature of proportional/prevalence data including how its variance and hence study's weight is calculated), we have used the double arcsine (freeman-Tukey) transformation in our meta-analysis as it is the recommended transformation method [72, 73]; this transformation overcomes both issues related to using the normal meta-analysis approach on untransformed prevalence data with the first issue being the problem of estimating a confidence interval that falls outside the 0–1 range, and the second issue of over-estimation of weights for studies with prevalence estimates that are at the extreme ends of zero to one [72]. We have used the *metaprop* command in STATA to conduct this double arcsine meta-analysis [74].

However, we are aware that this review was not without limitations. During the screening process, a significant number of studies have been excluded as they did not meet the inclusion criteria. The majority of the excluded studies included non-lab confirmed patients with COVID-19, therefore, bacterial coinfection and antibiotic use may be under- or over-reported. Disproportionate representation from North America and lack of eligible studies from regions other than Europe and Asia can also limit the generalizability of the results to other regions impacted by COVID-19, hence makes it difficult to make any conclusion about regional differences/variations; however, it is worth noting that the latter was not the objective of our study but rather we conducted a sub-group analysis by regions in order to explore the source of heterogeneity and as a sensitivity analysis to assess the sensitivity of the pooled estimates to the studies' geographical location. Additionally, the majority of studies included were conducted within the first 6 month of pandemic. Consequently, data included might not be up to date, which again, can compromise the generalizability of the results. In addition, the majority of studies included in the meta-analyses were retrospective studies with their inherently associated bias and limitations.

Alongside this, determining the appropriateness and justifiable need of antibiotic therapy, which is likely to be higher in prospective studies in comparison to retrospective studies, was not possible, as studies have mainly reported the number of patients prescribed antibiotics. Information such as indications, initiation timing and duration of antibiotic could assist in determining future appropriateness. Diagnostic tests and measures used to determine bacteriological infections were also under-reported. This is crucial to determine whether the infection is a true infection or bacterial colonisation.

High heterogeneity was reported across all meta-analyses, which warrants caution and conservative interpretation of the results. Attempts have been made to explore the source of heterogeneity through sub-group analyses based on regions and study design, in addition, the exclusion of the two studies (Russell *et al*. [44] and Puzniak *et al*. [34]) with the highest population, all of which have yielded similar high heterogeneity. We believe this can be attributed to

between -study variations such, how COVID-19 is diagnosed, definition of co-infection in each study, documenting of antibiotic use etc. Furthermore, it is worth mentioning that heterogeneity (measured by $I^2$) is often high in proportional/prevalence meta-analysis studies representing either false heterogeneity (resulted from the nature of proportional data in that even with small sample size studies, small variance could be observed) or true heterogeneity (resulted from true differences in prevalence estimates due to variations in the time points and places where these prevalence estimates were measured in each individual studies [73]. In addition, clear overlapping of the confidence intervals can be observed, despite the sub-group analysis highlighting some difference in the point estimates, however, the overlapping between the studies demonstrates a not statistically significant result indicating consistent results (i.e., no clear difference exist) among the various sub-groups.

A clear asymmetry is observed in the funnel plot generated for antibiotic use (*Fig 8*) which could be attributed to publication and/or heterogeneity of antibiotic use/prescription practices, nonetheless, this asymmetry was not statistically significant based on the P-value (0.59) of the Eggers test. However, results from funnel plot and Eggers test should be interpreted with cations in proportional/prevalence meta-analysis because these tests were originally developed for comparative/intervention meta-analysis data with the assumption that studies with positive results are more likely to be published compared to those with negative results but this assumption is not necessarily applicable and true for prevalence studies [73].

The inclusion of 3 (of the 22 studies included) non-peer reviewed studies might raise concerns regarding their quality [50, 51, 53]. However, two of these studies are now published, so it is unlikely to be of low quality [51, 53]. The remaining one, despite not being published, have still attained a "good" quality rating using the NOS, in addition, the study's weight in the forest plot is small, and therefore unlikely to affect the overall results.

Future reviews and studies should aim at diversifying study regions, and to include or conduct studies that are more up to date. Studies should also include data on the appropriateness of antibiotic therapy, diagnostic tests and measures used to determine the infection. However, despite these limitations, we believe the findings give good guidance regarding the need to improve the rationality of antibiotic prescribing in patients with COVID-19 to reduce the occurrence of AMR within facilities.

## Conclusion

This study demonstrates that the prevalence of bacterial coinfection amongst patients with COVID-19 was low, 5.62%, nevertheless, antibiotics use amongst COVID-19 patients was high (61.77%). However, the outcomes of this manuscript need to be interpreted with caution. Despite reporting low bacterial coinfection with the variability of the rate ranging between 2–10% amongst patients with COVID-19, when deciding to prescribe antibiotics to a patient, the difference between 2 and 10% prevalence would not be considered significant to most clinicians, and if antibiotic administration is delayed in patient with bacterial coinfection, it could result in poor prognosis. The findings of this study encourages a more rational approach to antibiotics prescribing in COVID-19 patients, an approach based on laboratory-confirmed diagnosis of coinfection, rather than clinical, advocating for more antimicrobial stewardship (AMS).

## Supporting information

**S1 Checklist. PRISMA 2020 checklist.**
(DOCX)

**S2 Checklist. PRISMA 2020 for abstracts checklist.**
(DOCX)

**S1 Data.**
(DOCX)

## Author Contributions

**Conceptualization:** Faisal Salman Alshaikh, Brian Godman, R. Andrew Seaton, Amanj Kurdi.

**Data curation:** Faisal Salman Alshaikh.

**Formal analysis:** Faisal Salman Alshaikh.

**Investigation:** Faisal Salman Alshaikh.

**Methodology:** Faisal Salman Alshaikh.

**Supervision:** Amanj Kurdi.

**Validation:** Brian Godman, Oula Nawaf Sindi, R. Andrew Seaton, Amanj Kurdi.

**Writing – original draft:** Faisal Salman Alshaikh.

**Writing – review & editing:** Faisal Salman Alshaikh, Brian Godman, Oula Nawaf Sindi, R. Andrew Seaton, Amanj Kurdi.

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
