## [Decision Letter · Decision Letter 0]

2 May 2022

PONE-D-22-06815Prevalence of Bacterial Coinfection and Patterns of Antibiotics Prescribing in Patients with COVID-19: A Systematic review and Meta-AnalysisPLOS ONE

Dear Dr. Alshaikh,

Thank you for submitting your manuscript to PLOS ONE. After careful consideration, we feel that it has merit but does not fully meet PLOS ONE’s publication criteria as it currently stands. Therefore, we invite you to submit a revised version of the manuscript that addresses the points raised during the review process.

We look forward to receiving your revised manuscript.

Kind regards,

Eili Y. Klein, PhD

Academic Editor

PLOS ONE

Journal Requirements:

4. We note that this manuscript is a systematic review or meta-analysis; our author guidelines therefore require that you use PRISMA guidance to help improve reporting quality of this type of study. Please upload copies of the completed PRISMA checklist as Supporting Information with a file name “PRISMA checklist”.

Additional Editor Comments:

This is an interesting and timely paper, however the reviewers have raised some concerns, so I am returning this for revision. The major concerns are around the descriptions of bias and the heterogeneity in the underlying studies. One reviewer raised significant concerns about the methodology, so please justify the methods, or add additional information about the heterogeneity and consider some sub-analyses to ensure the results are consistent. Additionally, in line with the sub-analyses, greater justification of regional breakdowns is needed, it is not clear why they would be broken down that way rather than by study design. The first reviewer raises several points that are worth addressing regarding this heterogeneity in terms of how each study was performed in looking for bacterial pathogens. Adding descriptions of the methods within each study, and potential grouping by similar studies, would go a long way to address these concerns. Both reviewers raised some concerns about the biases, so please add a bit more of an explanation about the issues there. Additionally, I would suggest stratifying the results by risk of bias in an additional sub-analysis.

Reviewers' comments:

Reviewer's Responses to Questions

**Comments to the Author**

1. Is the manuscript technically sound, and do the data support the conclusions?

Reviewer #1: Yes

Reviewer #2: No

2. Has the statistical analysis been performed appropriately and rigorously? 

Reviewer #1: I Don't Know

Reviewer #2: No

3. Have the authors made all data underlying the findings in their manuscript fully available?

Reviewer #1: Yes

Reviewer #2: Yes

4. Is the manuscript presented in an intelligible fashion and written in standard English?

Reviewer #1: Yes

Reviewer #2: Yes

5. Review Comments to the Author

Reviewer #1: Prevalence of Bacterial Coinfection and Patterns of Antibiotics Prescribing in Patients with COVID-19: A Systematic Review and Meta-Analysis.

Manuscript Number: PONE-D-22-06815

Summary

The topic of this manuscript is very important. The course of the COVID-19 pandemic remains uncertain and seems increasingly likely it will continue to impact medical systems into the future. What is more certain is the pandemic of antimicrobial resistance (ref). All use of antibiotics contributes to this pandemic, particularly antibiotic use in patients without a bacterial infection. Bacterial coinfection is a real, but very difficult to diagnose complication of viral pneumonia. For this reason, clinicians struggle to determine which patients are likely to benefit from antibiotics. Establishing baseline risk factors for bacterial coinfection in COVID-19 is critical in aiding clinicians to utilize antibiotics in this population. A systematic review and meta-analysis is an excellent tool to determine that risk. The authors perform a systematic review and meta-analysis of the available literature on coinfection in COVID-19. They find that coinfection occurs in about 5% of patients within 48 hours of COVID-19 diagnosis. They also find that antibiotic prescription is over 60% in that same population. They conclude that antibiotics are over prescribed for coinfection in COVID-19.

Major Comments:

While the concept of this study is very important for clinicians and antibiotic stewards, this not unique and is somewhat behind the literature. Multiple meta-analyses and systematic reviews, including one in PLOS ONE, have already been published with findings comparable to this manuscript. In response to these earlier studies, national guidelines statements on antibiotic use in COVID-19 have been published. Currently, clinicians and stewardship programs are advocating more restricted antibiotic use in COVID-19. While these studies have been done, this manuscript is the largest meta-analysis that this reviewer has come across which would serve to cement the literature on incidence of coinfection in COVID-19, validate ongoing antimicrobial stewardship efforts, and affirm clinician’s choices to use less antibiotics in COVID-19.

The authors stratified coinfection and antibiotic use by location. I have not previously seen much literature to suggest the pathophysiology of COVID-19 and coinfection are different between regions. Some regions have different baseline population ages and prevalence of comorbidities which can affect disease severity, and indirectly, risk for coinfection or antibiotic use. What is likely different between regions is robustness of diagnostic modalities for coinfection, availability of various antibiotics, and prevalence of antimicrobial stewardship programs. While some differences exist between the regions, the general message of relatively uncommon coinfection and antibiotic over utilization remains similar between groups. The major weakness of this regional analysis is significant bias towards the United States and Europe. The data are sparce from Asian locations, and there are no studies from Africa or South America. It is difficult to make any conclusions about regional differences given large lack of data from other geographic regions.

Three terms were used in the search. All of these terms are reasonable. This reviewer wonders if they may have picked up more literature if “bacterial,” or “super-infection” were used.

Studies performed in children were not included. It is likely few studies were preformed in children given COVID-19 is less morbid in children. None-the-less, inclusion might have been useful as children are often overlooked in the medical literature and left behind by evidence based medicine.

Non-peer reviewed publications were included in this analysis. This reviewer finds inclusion of non-peer reviewed manuscripts problematic. The rapid push to describe COVID-19 and publish experiences during the pandemic has resulted in many pre-print and non-peer reviewed articles being disseminated. These premature disseminations have resulted in retractions and risk undermining public trust of the medical literature. The authors appropriately list this as a potential limitation of the study later in the manuscript.

The authors chose to only include coinfection occurring within 48 hours of COVID-19 diagnosis. This is helpful as hospital acquired pneumonia occurs after prolonged hospital exposure and is pathophysiologically distinct from community acquired pneumonia.

Diagnosing coinfection is difficult and often done heterogeneously. Diagnosis can be based on clinical factors such as oxygenation, radiography, fever, physical exam findings. Diagnosis can also be aided with laboratory and microbiologic testing, such as inflammatory markers, cultures, and PCR. This is much more difficult in the setting of a pre-existing infection, which can mimic bacterial infection. It would be nice to see a description of how each study defined coinfection. This might possibly explain the high degree of heterogeneity between coinfection rates.

Figure 2 is an easy to read figure clearly demonstrating the finding of a roughly 5% coinfection rate, ranging from 2-10%. This is the figure that most readers will remember as the principle finding of the meta-analysis. This coinfection rate is consistent with rates described by other groups. The variability in rate is understandable given the heterogeneity of the study populations and coinfection diagnosis practices. While the mathematical variability within the 95% confidence interval (2 and 10%) is real, this degree of variability is clinically smaller. When deciding to prescribe antibiotics to a patient, the difference between 2 and 10% prevalence would not be considered significant to most clinicians.

The prevailing organisms causing coinfection are known common causes of pneumonia. It is surprising that E coli was the second most commonly identified organism though. Perhaps this is driven by studies including urinary tract infection (UTI) among their coinfections. If UTIs are driving this finding, it is worth mentioning by the authors, as E coli is an uncommon cause of community acquired pneumonia.

Studying antibiotic use adjacent to coinfection rates is a natural comparison that underlies the importance of determining the true risk of coinfection in COVID-19. It also emphasizes the critical need to effectively identify patients with coinfection and appropriately administer antibiotics. This reviewer is not surprised they found considerable heterogeneity of antibiotic prescribing practices for a number of reasons. Antimicrobial stewardship is performed heterogeneously between hospitals. It is likely that antimicrobial prescription practices were lower in studies performed at hospitals with robust antimicrobial stewardship programs.

funnel plot for antibiotic prescribing practices does not have ideal symmetry. This asymmetry could be attributed to publication bias or heterogeneity of antibiotic prescription practices. A comment to this effect may be warranted.

The authors perform a brief discussion of some of the high impact or outlier studies included in the analysis. This reviewer appreciates this discussion as it lends incite in to the design of included studies. It also lends incite into the high degree of heterogeneity of coinfection literature. One major factor in potential over diagnosis of coinfection is diagnostic culture of non-sterile body sites. The authors comment directly on this challenge in regards to colonization of the urinary track in older diabetic patients as well as colonization of the airways of mechanically ventilated patients.

The authors suggest that there is insufficient evidence for high coinfection rates to justify considerable empiric antibiotic prescribing. This reviewer believes that this manuscript is unable to completely support this statement. While coinfection rates are low, there is a proportion of patients who do have coinfection. It is very likely those patients would have a much worse prognosis if antibiotic administration was delayed. This reviewer takes this manuscript as evidence we need to better identify patients who are unlikely to have a coinfection and avoid using antibiotics in them. Given there is a (small) rate of coinfection, antibiotic use will continue to be necessary in COVID-19, the key is identifying those few patients and sparing antibiotics in the rest.

This study provides the most complete and thorough review and meta-analysis of coinfection and antibiotic prescription practices in COVID-19 to date. Though, the message of the review is not dissimilar to previous systemic reviews and meta-analyses on the topic. That said, the message is abundantly clear, coinfection is a relatively uncommon complication of COVID-19, despite this, antibiotic prescription is very high. This manuscript serves to bolster the ongoing efforts to reduce unnecessary antibiotic prescribing in COVID-19. This study will no doubt serve to support further studies into the timely identification of patients with coinfection and striking an appropriate balance between antibiotic prescription and reducing morbidity and mortality in bacterial coinfections.

Minor Comments:

Line 100: Categorizing information regarding hydroxychloroquine use as “misinformation” strikes this reviewer as problematic and potentially political. There is a growing body of empirical evidence on hydroxychloroquine use in COVID-19. The evidence indicates hydroxychloroquine is not beneficial and likely harmful. Regardless, this study is not the appropriate position to determine what information should be considered “misinformation.”

Line 184: This reviewer is not a trained statistician and unable to comment on the appropriateness of the statistical methods.

Figure 1: Figure is clean and makes understanding study inclusion/exclusion simple.

Table 1: Table is easy to read and compare study similarities and differences.

Figure 3: Sub stratification by region is an interesting aspect of this analysis, this figure demonstrates some of the subtleties between the selected regions

Figure 4: stratifying the findings by study design is good, it shows small benefit of the superior prospective studies, but doesn’t change the message of the manuscript by much.

Figure 5: This figure will but just behind figure 2, in memorability for readers. It contrasts nicely with figure 2, drawing stark contrast between coinfections and antibiotic use.

Line 229: nearly 90% of the patients included in analysis were from two studies. It is likely these studies bear outsized weight on the final outcome of the analysis. The authors recognize this outsized contribution appropriately.

Line 402-406: This reviewer agrees with the hypothesis that coinfection rates are likely linked to diagnostic practice patterns at various hospitals.

Line 420-427: This reviewer agrees with the hypothesis that study design likely influenced coinfection rates owing to more rigorous coinfection definitions in prospective studies.

Reviewer #2: This manuscript was a systematic review and metanalysis (SRMA) of the prevalence of bacterial co-infections and use of antibiotics in hospitalized patients with COVID-19. Similar to prior SRMAs, the authors describe relatively low prevalence of co-infection and high antibiotic use.

The methods used in the meta-analysis need to be further examined. The authors used a random-effects model, however the within-study and between-study variance (Tau squared statistic) is not reported (both of which influence study weight). As is currently reported for all analyses in this manuscript, the weights of each included study appears to be nearly equal (all in the 4-6% range). However the authors acknowledge (lines 229-231) that two studies made up 86.5% of the total population. Therefore it is not clear how these two studies appear to be weighted nearly equally as the other included studies. Is this the correct weight distribution for these random effects models? If this is statistically accurate, mention should be made of this.

Another methodological concern is the extent to which the authors assess risk of biases in this SRMA. Publication bias was assessed using Funnel plots and Egger’s asymmetry test. The authors conclude there is no publication bias, however Figure 8B appears to be quite asymmetric with numerous studies laying outside of the funnel lines. This is confusing. It is also concerning that it does not appear that other biases were considered or assessed. For example, many SRMAs will have an entire section of the manuscript for a risk of bias assessment. If the included studies are not examined for different biases, how is it known that these biases are not reflected in the SRMA? The Cochrane Collaboration has a Risk of Bias tool, which could serve as an example. A risk of bias assessment is also part of the PRISMA checklist. Additionally many SRMAs would make their PRISMA checklist available as supplemental information, which was not done here.

All of the meta-analyses included had exceedingly high levels of heterogeneity (I2). The heterogeneity is listed in the results section, but this needs to be discussed in the discussion of the paper. It is very difficult to truly draw conclusions with such high degrees of heterogeneity. Perhaps this could speak to some of the methodological concerns addressed above, or the true heterogeneity of the included studies. Additionally, one way to explore heterogeneity may be with subgroup analyses. For example, eliminating the two studies with the highest populations (and therefore possibly the highest weights), or studies with large within-study variance.

Numerous results of sub-group analyses (retrospective vs prospective, region-specific) are discussed in the results and extensively in the discussion. However, it doesn’t appear as there are truly any clear differences, as all of the confidence intervals for the sub-groups overlap. This is not brought up in the discussion, and is also ignored when attempting to draw conclusions that differences exist.

Additional comments:

- Please spell out “COVID-19” the first time it is used (line 88)

- Numerous times throughout the manuscript the phrase “COVID-19 patients” is used. This should be replaced with “patients with COVID-19”.

- Parts of the introduction (lines 91-98) do not seem relevant to the current study, and it is detracting from the manuscript.

- Clarify in the inclusion criteria for antibiotic use, at what time this is (line 173). Is this antibiotics within the first 48 hours?

- In the methods, please clarify if trial/study characteristics and outcomes were all pre-specified

- In all figures, please better label information in the figure itself. Spell out or clarify what “ES” is (effect size?). is this just proportion? It may be more clear if labeled that way. It is also often helpful to have the population of each study included

- Figures 4 and 6, the I2 statistic and p-value are missing for the prospective cohort sub-analysis.

- Address line 329

- Consider re-wording lines 337-338 as “The aim of this systematic review and meta-analysis was to determine the prevalence of bacterial coinfection and antibiotic use in patients with COVID-19.”

- Line 344 should read “the findings in this review are consistent…”

- Line 346 should read “Bacterial coinfection prevalence was low…”

- Lines 366-368: Was this study looking at co-infection (i.e., first 48 h) or hospital-acquired/nosocomial bacterial coinfection?

- It is not clear why “new variants, updated treatment regimens, and variations in measures for SARS-CoV-2 testing” would impact bacterial co-infection (lines 456-458)

- Consider removing “so it is unlikely to be of low quality” (line 472)

- The argument in lines 472-474 does not make sense, as discussed above, all studies were weighted nearly equally.

- Consider rephrasing the paragraph in lines 476-481. A systematic review is limited by the available data published, which answers your questions developed a priori.

- The last sentence in the conclusion (lines 489-491) seems out of place and not as relevant to the findings of this study.

6. PLOS authors have the option to publish the peer review history of their article (what does this mean?). If published, this will include your full peer review and any attached files.

Reviewer #1: No

Reviewer #2: No

---

## [Author Response · Author response to Decision Letter 0]

18 Jun 2022

Reviewer #1

The major weakness of this regional analysis is significant bias towards the United States and Europe. The data are sparse from Asian locations, and there are no studies from Africa or South America. It is difficult to make any conclusions about regional differences given large lack of data from other geographic regions.

[Reply] Thank you for this comment. We do acknowledge this, however, during the data collection phase, relevant data from other regions were not available at the time being, thus further studies in lacking regions need to be conducted in to understand regional differences. Furthermore, we want to clarify that making conclusions about regional differences was not the main objective of our study but rather we conducted a sub-group analysis by regions in order to explore the source of heterogeneity and as a sensitivity analysis to assess the sensitivity of the pooled estimates to the studies’ geographical location.

We have now added a statement to the “Strengths and Limitations” section to reflect this. We hope that this is acceptable. 

Three terms were used in the search. All of these terms are reasonable. This reviewer wonders if they may have picked up more literature if “bacterial,” or “super-infection” were used

[Reply] Thank you for your comment. In regards, to the use of the term “Bacterial”, we planned so start broad in our key search strategy to include as much literature as possible, therefore the term “Coinfection” was used, and later narrowed it down to bacterial infection, kindly refer to Appendix 1

In regards to not including the term “Superinfection”, we would like to clarify that we did not include superinfection because it implies that the second infection is superimposed on an earlier one that it being treated with an antibacterial agent; our systematic review meta-analysis, however, was primarily focused on co-infections occurring within 48 hours of confirmed SAR-CoV-2 infection. We hope this is OK with you.

Studies performed in children were not included. It is likely few studies were performed in children given COVID-19 is less morbid in children. None-the-less, inclusion might have been useful as children are often overlooked in the medical literature and left behind by evidence based medicine.

[Reply] Thank you for your comment. We acknowledge this, but because children and adult have different pathophysiology and could well have different antibiotic use patterns, we believe it would not have been relevant to combine these two populations together. In addition, there was a potentially low prevalence of COVID 19 in children during earlier waves of the pandemic with low morbidity and mortality. We have now added this into the paper. We accept this is beginning to change alongside equal concerns regarding the over use of antimicrobials in this population (some of the co-authors have recently published on this in Bangladesh with a study on the situation in India recently submitted). Consequently a similar study using children population is now being planned. We hope this is acceptable

Non-peer reviewed publications were included in this analysis. This reviewer finds inclusion of non-peer reviewed manuscripts problematic. The rapid push to describe COVID-19 and publish experiences during the pandemic has resulted in many pre-print and non-peer reviewed articles being disseminated. These premature disseminations have resulted in retractions and risk undermining public trust of the medical literature. The authors appropriately list this as a potential limitation of the study later in the manuscript.

[Reply] Thank you for your comment. Although at the time of the analysis only 3 of the included studies were unpublished,, now two of these have become published, the exception being Crotty MP et al 2020. Nevertheless, to address the reviewer’s comments, we have conducted a sensitivity analysis excluding the non-peer reviewed studies and the results remained consistent. We hope this is acceptable. 

Furthermore, we would like to highlight that this approach of including non-peer reviewed publications in such meta-analysis during the covid-19 pandemic has been a common practice with further sub-group analysis based on peer-review status which is what we have done in our current study as well. An example of a previous systematic review/meta-analysis which used this approach is the study by Kurdi A et al, 2020 “A systematic review and meta-analysis of the use of renin angiotensin system drugs and COVID-19 clinical outcomes: What is the evidence so far?” We have now added this rationale into the Methodology, and hope this is now acceptable.

The authors chose to only include coinfection occurring within 48 hours of COVID-19 diagnosis. This is helpful as hospital acquired pneumonia occurs after prolonged hospital exposure and is pathophysiologically distinct from community acquired pneumonia

[Reply] We thank the reviewer for this valuable comment. 

Figure 2 is an easy to read figure clearly demonstrating the finding of a roughly 5% coinfection rate, ranging from 2-10%. This is the figure that most readers will remember as the principle finding of the meta-analysis. This coinfection rate is consistent with rates described by other groups. The variability in rate is understandable given the heterogeneity of the study populations and coinfection diagnosis practices. While the mathematical variability within the 95% confidence interval (2 and 10%) is real, this degree of variability is clinically smaller. When deciding to prescribe antibiotics to a patient, the difference between 2 and 10% prevalence would not be considered significant to most clinicians.

[Reply] We thank the reviewer for this valuable comment, this has now been acknowledged by the authors in the conclusion section.

The prevailing organisms causing coinfection are known common causes of pneumonia. It is surprising that E coli was the second most commonly identified organism though. Perhaps this is driven by studies including urinary tract infection (UTI) among their coinfections. If UTIs are driving this finding, it is worth mentioning by the authors, as E coli is an uncommon cause of community acquired pneumonia.

[Reply] Thank you for your comment. We have now added a statement in the discussion to address the reviewer’s comment. 

Funnel plot for antibiotic prescribing practices does not have ideal symmetry. This asymmetry could be attributed to publication bias or heterogeneity of antibiotic prescription practices. A comment to this effect may be warranted.

[Reply] Thank you for your comment. We have now added a paragraph in the limitations section to address the reviewer’s concerns. We hope that this is acceptable now. 

The authors perform a brief discussion of some of the high impact or outlier studies included in the analysis. This reviewer appreciates this discussion as it lends incite in to the design of included studies. It also lends insight into the high degree of heterogeneity of coinfection literature. One major factor in potential over diagnosis of coinfection is diagnostic culture of non-sterile body sites. The authors comment directly on this challenge in regards to colonization of the urinary track in older diabetic patients as well as colonization of the airways of mechanically ventilated patients.

[Reply] We thank the reviewer for this valuable comment.

The authors suggest that there is insufficient evidence for high coinfection rates to justify considerable empiric antibiotic prescribing. This reviewer believes that this manuscript is unable to completely support this statement. While coinfection rates are low, there is a proportion of patients who do have coinfection. It is very likely those patients would have a much worse prognosis if antibiotic administration was delayed.

[Reply] Thank you for your comment, this has now been elaborated in the conclusion section. We hope that this is acceptable now.

This reviewer takes this manuscript as evidence we need to better identify patients who are unlikely to have a coinfection and avoid using antibiotics in them. Given there is a (small) rate of coinfection, antibiotic use will continue to be necessary in COVID-19, the key is identifying those few patients and sparing antibiotics in the rest.

[Reply] Thank you for highlighting this important point, the authors agree with the reviewer’s comment. 

This study provides the most complete and thorough review and meta-analysis of coinfection and antibiotic prescription practices in COVID-19 to date. Though, the message of the review is not dissimilar to previous systemic reviews and meta-analyses on the topic. That said, the message is abundantly clear, coinfection is a relatively uncommon complication of COVID-19, despite this, antibiotic prescription is very high. This manuscript serves to bolster the ongoing efforts to reduce unnecessary antibiotic prescribing in COVID-19. This study will no doubt serve to support further studies into the timely identification of patients with coinfection and striking an appropriate balance between antibiotic prescription and reducing morbidity and mortality in bacterial coinfections.

[Reply] The authors thank the reviewer for his positive overall feedback on this manuscript.

Minor comments

[Reply] The authors thank the reviewer for his comments, all relevant comments which needs addressing have been addressed accordingly 

 

Reviewer #2

The methods used in the meta-analysis need to be further examined. The authors used a random-effects model, however the within-study and between-study variance (Tau squared statistic) is not reported (both of which influence study weight

[Reply] Thank you for your comment, the Tau Squared Value for both meta-analysis have now been mentioned in the results section, please review lines (240-241 & 252) 

As is currently reported for all analyses in this manuscript, the weights of each included study appears to be nearly equal (all in the 4-6% range). However the authors acknowledge (lines 229-231) that two studies made up 86.5% of the total population. Therefore it is not clear how these two studies appear to be weighted nearly equally as the other included studies. Is this the correct weight distribution for these random effects models? If this is statistically accurate, mention should be made of this.

[Reply] Thank you for your comment. We would like to clarify that the variance of proportional/prevalence data; hence, the given weight for each study, unlike the variance of association measures such as odds ratio, is determined not only by the sample size but also the estimated prevalence; this means that pooling prevalence data via the normal meta-analysis approach would violate the statistical validity of the pooled estimates because the variance for studies with a prevalence estimate value at the extreme ends of zero to one would be toward zero which in turn results in artificial inflation of the estimated weights for these studies in the final pooled prevalence estimate. Therefore, transformation of the prevalence data is required. Accordingly, we have used the double arcsine (freeman-Tukey) transformation in our meta-analysis as it is the recommended transformation method. Therefore, the weights of studies with large sample size might be similar to the weight of much smaller studies because even with small studies, little variance still be observed resulting is much larger weight for these smaller studies; in fact this has been observed in other published prevalence meta-analysis studies whereby the given weights for large and smaller studies were comparable as in the study by Munn, Z et. al (2015). 

We have now included a statement in the discussion to reflect this. We hope that this is acceptable. 

Another methodological concern is the extent to which the authors assess risk of biases in this SRMA. Publication bias was assessed using Funnel plots and Egger’s asymmetry test. The authors conclude there is no publication bias, however Figure 8B appears to be quite asymmetric with numerous studies laying outside of the funnel lines. This is confusing. It is also concerning that it does not appear that other biases were considered or assessed. For example, many SRMAs will have an entire section of the manuscript for a risk of bias assessment. If the included studies are not examined for different biases, how is it known that these biases are not reflected in the SRMA?

[Reply] Thank you for your comment, we acknowledge that the funnel plot looks asymmetrical, however this was not statically significant from the eggers test. Publication bias is bias assessment, and risk of bias is similar to quality assessment (potential confusion), therefore, table 2, has been added, table 2 illustrates the risk of bias assessment using Newcastle-Ontario Scale (NOS) that has been undertaken in this study. Furthermore, regarding Funnel plots and Egger’s asymmetry test (as we explained in our reply to reviewer one above), we would like to mention that results from funnel plot and Eggers test should be interpreted with cations in proportional/prevalence meta-analysis because these tests were originally developed for comparative/intervention meta-analysis data with the assumption that studies with positive results are more likely to be published compared to those with negative results but this assumption is not necessarily applicable and true for prevalence studies

We have now added Table 2 and a statement in the discussion to reflect these changes. We hope that this is now acceptable.

All of the meta-analyses included had exceedingly high levels of heterogeneity (I2). The heterogeneity is listed in the results section, but this needs to be discussed in the discussion of the paper. It is very difficult to truly draw conclusions with such high degrees of heterogeneity. Perhaps this could speak to some of the methodological concerns addressed above, or the true heterogeneity of the included studies. Additionally, one way to explore heterogeneity may be with subgroup analyses. For example, eliminating the two studies with the highest populations (and therefore possibly the highest weights), or studies with large within-study variance

[Reply] Thank you for your comment. In terms of the high levels of heterogeneity, we would like to mention that it is worth mentioning that heterogeneity (measured by I2) is often high in proportional/prevalence meta-analysis studies representing either false heterogeneity (resulted from the nature of proportional data in that even with small sample size studies, small variance could be observed) or true heterogeneity (resulted from true differences in prevalence estimates due to variations in the time points and places where these prevalence estimates were measured in each individual studies as well as variations in the definitions of COVID-19 diagnosis, bacterial con-infections etc. Furthermore, as recommended by the reviewer, we have conducted further sub-group analysis by excluding the two studies with the highest population and the pooled estimates were comparable

We have now revised the manuscript to reflect the above changes. We hope that this is acceptable now.

Numerous results of sub-group analyses (retrospective vs prospective, region-specific) are discussed in the results and extensively in the discussion. However, it doesn’t appear as there are truly any clear differences, as all of the confidence intervals for the sub-groups overlap. This is not brought up in the discussion, and is also ignored when attempting to draw conclusions that differences exist.

[Reply] Thank you for your comment. We have now added a paragraph in the discussion section to reflect this. We hope that this is acceptable now. 

Minor Comments Sections

[Reply] The authors thank the reviewer for his comments, all relevant comments which needs addressing have been addressed accordingly, with the exception of the following:

• In all figures, please better label information in the figure itself. Spell out or clarify what “ES” is (effect size?). Is this just proportion? It may be clearer if labelled that way. It is also often helpful to have the population of each study included

[Reply] Thank you for this comment. All graphs are generated automatically by STATA, only certain components of the graph can be modified by the user, unfortunately ES (Effect Size) is automatically generated as an acronym by STATA, however, to address this concern we have spelled out ES in line (244) to make it clearer for future readers.

• Figures 4 and 6, the I2 statistic and p-value are missing for the prospective cohort sub-analysis.

[Reply] Thank you for this comment. This has now been addressed and the plots have been modified

---

## [Editor Report · Decision Letter 1]

19 Jul 2022

Prevalence of Bacterial Coinfection and Patterns of Antibiotics Prescribing in Patients with COVID-19: A Systematic review and Meta-Analysis

PONE-D-22-06815R1

Dear Dr. Alshaikh,

We’re pleased to inform you that your manuscript has been judged scientifically suitable for publication and will be formally accepted for publication once it meets all outstanding technical requirements.

Kind regards,

Eili Y. Klein, PhD

Academic Editor

PLOS ONE

Additional Editor Comments (optional):

I believe that the responses were adequate and no additional revisions are needed.
---

## [Editor Report · Acceptance letter]

22 Jul 2022

PONE-D-22-06815R1 

Prevalence of Bacterial Coinfection and Patterns of Antibiotics Prescribing in Patients with COVID-19: A Systematic review and Meta-Analysis 

Dear Dr. Alshaikh:

I'm pleased to inform you that your manuscript has been deemed suitable for publication in PLOS ONE. Congratulations! Your manuscript is now with our production department. 

Kind regards, 

on behalf of

Dr. Eili Y. Klein 

Academic Editor

PLOS ONE